# Can Isogroup Selection of Highly Zoophagous Lines of a Zoophytophagous Bug Improve Biocontrol of Spider Mites in Apple Orchards?

**DOI:** 10.3390/insects10090303

**Published:** 2019-09-18

**Authors:** François Dumont, Denis Réale, Éric Lucas

**Affiliations:** Département des Sciences Biologiques, Université du Québec à Montréal, CP 8888, Succ. Centre Ville, Montréal, QC H3C 3P8, Canada; reale.denis@uqam.ca (D.R.); lucas.eric@uqam.ca (É.L.)

**Keywords:** biological control, artificial selection, strain selection, isogroup lines, zoophytophagous predators, diet specialization, mullein bug

## Abstract

Zoophytophagous predators provide benefits in agroecosystems when feeding on pests, but they can also cause crop damage. Optimizing the use of zoophytophagous predators as biocontrol agents would require improving pest control and/or limiting damage. Populations of a zoophytophagous species can be composed of a mix of individuals diverging in their level of diet specialization. Consequently, depending on their level of zoophagy, individuals would vary widely in the benefits and risks they provide to pest management. We tested the hypothesis that manipulating the composition of the population of a zoophytophagous insect, the mullein bug, *Campylomma verbasci* (Hemiptera: Miridae), towards an increased zoophagy would increase their net benefit in an apple orchard. We compared the inherent benefits and risks of two different isogroup lines of mullein bug that genetically differed in their level of zoophagy. In spring, when damage occurs, both strains infrequently punctured apple fruit, which rarely lead to damage and therefore represented a low risk. During summer, only the highly-zoophagous line impacted the spider mite population, while the lowly-zoophagous line did not differ from the control treatments. We concluded that manipulating the composition of the zoophytophagous predator population provided extra net benefits that improved pest control.

## 1. Introduction

Zoophytophagous predators can substitute prey with plant food items to deal with prey shortage or to complement animal diet [1]. These predators are frequently encountered in numerous agricultural systems [2] and can reduce pest populations considerably [3]. Their capacity to substitute phytophagy for zoophagy allows them to survive in agricultural systems during prey shortage and to overcome rapid spatial and temporal changes in prey availability [1,4]. Some zoophytophagous predators are currently used as biological control agents. For instance, the release of the zoophytophagous predator *Nesidiocoris tenuis* (Reuter) (Hemiptera: Miridae) decreased a white fly *Bemisia tabaci* (Gennadius) (Hemiptera: Aleyrodidae) population by about 90% [5,6], and regulated populations of the South American tomato pinworm, *Tuta absoluta* (Meyrick) (Lepidoptera: Gelechiidae) [7,8]. In contrast, some zoophytophagous predators have an ambiguous status. Their phytophagous behaviour can cause more damage than the benefits provided by their predatory behavior. For instance, the beneficial role of the mullein bug *Campylomma verbasci* (Meyer) (Hemiptera: Miridae) in apple orchards has been neglected because of potential crop damage (i.e., this insect causes dark corky warts surrounded by a depression at the surface of the fruit when feeding on developing apple fruits) [9,10].

In North America, mullein bugs are well-known predators of two important apple tree pests, the European red spider mite *Panonychus ulmi* (Koch) (Acari: Tetranychidae) and the two-spotted spider mite *Tetranychus urticae* (Koch) [11,12]. Both red and two-spotted spider mites can cause massive damage late in the season during the pre-harvest period, such as reduction of apple fruit size, downgrading, and crop losses the following season [13]. Mullein bug nymphs are the first predators to feed on red spider mites in spring. The first generation emerges early in the growing season and is synchronized with both apple tree bloom and the first generation of red spider mites [12,14,15]. Early season predators usually have a considerable impact on pest populations for the whole production season [16,17,18]. Moreover, the beneficial effect of mullein bugs can last for the whole summer, as two or three generations can be observed during the growing season [11]. Damage, though, does not occur once fruit diameter [19] becomes larger than ~10–13 mm [19,20]. Therefore, only nymphs of the first generation cause damage to apples [9,19,21]. The occurrence of damage is strongly linked to pest (aphid and mite) density and, to a lesser extent, intraguild predator presence [10]. Since damage on apple fruits is restricted to a short period (about two or three weeks) and benefits can extend over the entire apple production period, it could be interesting to consider methods that exploit the potential of the mullein bug as a biocontrol agent.

Recently, we reported genetic variation in foraging behavior of the zoophytophagous mullein bug [22,23,24]. These genetic differences included variation in the level of zoophagy (some lines killed significantly more spider mites than others) [22], cannibalism (highly-zoophagous lines are more cannibalistic than lowly-zoophagous one) [24], and food specialization for either animal or plant resources [23]. Various nutritional strategies coexist in mullein bug populations and can vary from highly-zoophagous genotypes (HZ: highly-zoophagous line) at one extreme, relying mainly on prey to meet their energy requirements, and at the other extreme, LZ genotypes (LZ: lowly-zoophagous line), preferentially feeding on plant resources, such as pollen. Consequently, some individuals are potentially more beneficial in crop production (i.e., those HZ genotypes) by improving the benefit/risk ratio related to the use of zoophytophagous predators [25].

In the present study, we evaluated whether a highly-zoophagous mullein bug line (HZ line) would provide more benefits in apple orchards than lowly-zoophagous lines (LZ line). Our first hypothesis is that the HZ line generates less damage to apples than the LZ line. In spring, we investigated whether there were differences in the number of punctures observed on apple fruitlets between an HZ and a LZ line in absence of prey. In these circumstances, mullein bugs are potential pests on apple trees [19], but diet specialization [23] could induce differences in the number of punctures. Secondly, we tested the hypothesis that HZ lines have a greater impact on spider mites than LZ lines by comparing the effect of mullein bugs on spider mite populations.

## 2. Materials and Methods

### 2.1. Isogroup Line Foundation

We used mullein bug nymphs from two isogroup lines selected from a group of twelve lines raised in our laboratory (Laboratoire de Lutte Biologique, Université du Québec à Montréal) [22,23]. The initial population was composed of individuals captured on apple trees and mullein plants, either as eggs (by cutting and collecting new branches, in December 2011) or as adults (in summer 2011 and 2012) in different apple orchards in Quebec. Captured individuals were reared in muslin cages containing two mullein plants, two soybean plants (*Glycine max*), and two potato plants (*Solanum tuberosum*). Green peach aphids *Myzus persicae* (Sulzer) (Hemiptera: Aphididae), two-spotted spider mites *T. urticae*, and mixed flower pollen were provided ad libitum. Large nymphs (5th instar) were individually kept in 10 cm diameter Petri dishes until adulthood (pollen, aphid, and spider mites were provided on a potato leaf inserted on agar gelatin). When these individuals reached adulthood, they were sexed and some were randomly selected to establish isogroup lines. Each isogroup line was initially composed of two virgin females and two males and maintained during nine generations (assuming a generation every 40 days). The isogroup lines was kept in an acrylic glass cage (30 × 30 × 30 cm) containing one mullein plant, one soybean plant, and two potato plants (with aphids, spider mites and pollen on it). In this series of experiments, we selected both the lines with the highest and the lowest level of zoophagy (hereafter referred to as HZ = highly-zoophagous and LZ = lowly-zoophagous line) on two-spotted spider mites (using the method outlined in Dumont et al., 2016).

The isogroup lines method consists of setting up a set of genetically different lines, which captures the genetic variance of a given trait [26]. The founder effect and genetic drift limit the genetic variance within each isogroup line [27]. Isogroup lines are kept under controlled identical conditions. Hence, the phenotypic variance among lines is mainly influenced by among-line genetic differences [26].

### 2.2. Mass Rearing

To produce a large number of individuals necessary for the experiments, HZ and LZ lines were reared in several muslin cages containing four potato and two soybeans plants, infested with aphids and spider mites, respectively. Mullein bugs were also provided ad libitum with pollen and a solution of sugar and distilled water. The cages were kept in a greenhouse at ~30 °C, ~60% relative humidity (R.H.), and 16 h of daylight.

### 2.3. Study Sites

The first two experiments in isogroup line comparison were run during summer 2013. Experiments on plant damage were run in an apple orchard located in Saint-Joseph-du-Lac (Laurentians, Quebec, Canada) (45°32′00′′ N; 74°00′00′′ W) and experiments on spider mite control took place in an orchard located in Rougemont (Monteregie, QC, Canada, 45°26′00′′ N; 73°03′00′′ W). We implemented the experiment on control of spider mites in a different orchard, because the first orchard was not infested by spider mites in August (a condition required to run this experiment). During both experiments, neither insecticides nor acaricides were used on the experimental trees. In 2014, all the experiments were implemented at the Rougemont orchard.

### 2.4. Experimental Design

#### 2.4.1. Damage on Apple Fruitlets

We compared damage caused by HZ and LZ lines on McIntosh apple trees (Kain and Agnello [19] observed that the mullein bugs caused damage to that variety) starting at bloom. On 18 May2013, forty muslin sleeve cages (20 × 20 × 70 cm) were installed on forty single branches with at least two flower clusters. Branches were previously inspected and cleaned of all arthropods. Flowers were manually pollinated with a fine paintbrush to ensure fruit development. Two young nymphs (L1 and/or L2) of either the HZ or the LZ line were introduced in each sleeve (i.e., 20 HZ and 20 LZ). Sleeves remained closed until the end of June (27 June 2013). On that day, apple fruitlets in each sleeve were collected. In the laboratory, fruitlets were thoroughly inspected with 40×binocular magnifiers and the number of punctures (damage) was counted. At this stage of apple fruit development, mullein bug punctures are characterized by a small circular depression on the surface of fruitlets [9]. Apple fruitlets were rated following Kain and Agnello [19]: fruit with a single puncture or less (0 or 1) (U.S. Fancy grade); two punctures or more (≥2) (downgraded). We inspected 54 and 52 fruitlets for the LZ and the HZ lines, respectively. An additional seven sleeve cages were installed as control treatment (without mullein bug nymphs). Nineteen apple fruitlets were retrieved in these sleeve cages.

#### 2.4.2. Spider Mite Predation

In the experiment on control of spider mites, we compared the efficiency of HZ vs. LZ lines to control for spider mite population growth in an apple orchard. Two independent experiments were done using similar set-ups. The first experiment was carried out using 15 cm-long muslin sleeve cages. In the second experiment, released individuals were not caged and were free to disperse. The sleeve-caged test indicates the maximum impact mullein bugs can have on a given limited location (i.e., a small apple branch). In contrast, the free-ranging test allows mullein bugs to disperse from the release points (and sampled branches), which provides the realistic efficiency of the predator in a complex environment.

Fourteen sets of three apple trees (Spartan cultivar), each separated by a buffer tree, were selected to run both experiments simultaneously. On each tree, four branches were selected (at major cardinal points): one was equipped with a sleeve cage and the three others were for the free-ranging test. Prior to setting up the sleeve cage, we removed any insect present on the branch, with the exception of spider mites.

The three treatments consisted of (1) a control, without addition of mullein bug nymphs, (2) a HZ line treatment, and (3) a LZ line treatment. In the sleeve cage tests, three nymphs (N3 to N5 stages) were introduced per sleeve cage, whereas in free-ranging tests, four nymphs (N3 to N5) were released on each experimental branch.

The populations of two-spotted and red spider mites were estimated on each branch by counting every mobile form of both species on the underside of five randomly selected leaves (three leaves for the sleeve-cage test). For the free-ranging test, the populations of spider mites were estimated on the day of introduction (day 0; 14 August 2013), on day 7 (21 August 2013), and on day 21 (4 September 2013). The estimation of spider mite populations for the sleeve-cage test was done on the day of introduction (day 0; 18 August 2013) and on day 14 (1 September 2013). Red spider mites were very scarce and were not considered in our analysis.

### 2.5. Statistical Analysis

For the experiment on damage, the probability of an apple fruitlet to be downgraded (≥2 mullein bug punctures) with respect to treatment (HZ or LZ line) was tested by implementing a generalized linear mixed model (GLMM) for binomial distribution (0 = ‘fancy’, 1 = downgraded). Some apples came from the same cluster, and thus this variable was included in the model as a random effect because of the lack of independence among apples from the same cluster. The proportion of clusters with damage was compared between treatments using a GLMM for a binomial-distributed response variable. For GLMM, we used the likelihood ratio test (LRT; α = 0.05) to determine the significance of fixed effects [28].

For the experiment on control of spider mites in sleeve cages, the two-spotted spider mite populations were compared among treatments (control without mullein bug, HZ, or LZ line) using linear mixed model (LMM) and Tukey’s test (using the *glht* function; package *multcomp*). The initial number of spider mites was added as a covariate (fixed effect) and the plot as a random effect. For the free-ranging test, a linear mixed model was run to test the number of spider mites after 7 and 21 days, according to the treatment and the initial population. Since three replicates were taken on each tree, the variable tree ID was included as a random effect in the model nested within the plot. We used the *drop1* function to obtain the *p*-value of each variable (this function drops each explanatory variable in turn and compares differences among models to a Chi-square distribution) [28]. All statistical analyses were run on R software [29].

## 3. Results

### 3.1. Damage on Apple Fruitlets

In the HZ line treatment, 94.4% (17 out of 18 fruitlets) of the apple fruitlets were classified as ‘fancy’ (one or no puncture), which did not differ from the 90.6% (29 out 32 fruitlets) in the LZ line treatment (LRT = 0.24; df = 1; *p* = 0.62). On the 19 apple fruitlets found in control treatment sleeves, no damage similar to a puncture by a mullein bug was found.

The proportions of clusters showing damage were not different between the HZ (36.8 ± 49.6%) and the LZ lines (55.6 ± 51.1%) (β = −0.76 ± 0.67; z = −1.14; *p* = 0.26).

### 3.2. Spider Mite Predation

In sleeve cages, the mean number of two-spotted spider mites per leaf was 21.7 (±10.8) on the day of mullein bugs introduction. Fourteen days later, the number of mites differed significantly among the three treatments, with an average of 89.9 (±58.85) mites/leaf in the control treatment, 48.29 (±34.05) mites/leaf in the HZ treatment, and 71.05 (±44.41) mites/leaf in the LZ treatment (likelihood ratio test (LRT) = 11.65; *p* = 0.003; Figure 1). Mites were significantly less abundant in the HZ treatment that in the control one, whereas the LZ treatment did not differ significantly with the HZ or the control treatment. The initial number of mites did not have an effect on the number of mites at the end of the test (LRT = 1.44; *p* = 0.23).

In the free-ranging test, the initial two-spotted spider mite population was 22.04 (±15.93 s.d.) individuals per leaf. Seven day later, we counted on average 28.39 (±15.89) mites/leaf in the control, 26.77 (±16.38) mites/leaf in the HZ, and 27.19 (±15.41) mites/leaf in the LZ treatment (Figure 2), which was not statistically different (LRT = 0.60; *p* = 0.74). The number of mites after 7 days depended on the initial mite population (LRT = 11.51; *p* = 0.0007).

At day 21, the average number of mites per leaf had increased to 100.82 (±42.31) in the control, to 81.38 (±25.71) in the HZ, and to 89.15 (±32.72) ind/leaf in the LZ treatment (Figure 2). The HZ line decreased significantly in the number of spider mites compared to the control treatment, but that was not the case for the LZ line (LRT = 6.54; *p* =0.038). The initial number of mites did not have an effect on the number of mites after 21 days (LRT = 0.04; *p* = 0.84).

## 4. Discussion

In agroecosystems, the benefits provided by zoophytophagous predators feeding on pests can be thwarted by the damage they cause by foraging on crops [30,31]. Biological control management programs promoting the use of zoophytophagous predators may increase the ratio of benefits/risks associated with such agents [31]. Genetic differences in foraging behavior/strategy provide an opportunity to increase benefits and/or to decrease risks either by artificially selecting traits of interest, or by agricultural practices favoring certain genotypes in a population of zoophytophagous insects [22,23]. Our results indicate that the highly-zoophagous mullein bug line (HZ) had a significant impact on two-spotted spider mite populations, whereas the lowly-zoophagous line (LZ) did not. The same trend was observed both in sleeve-caged and free-ranging tests. However, HZ and LZ lines did not differ significantly in their impact on spider mite populations. In our experiment, three to four mullein bug nymphs per branch were needed to significantly affect the number of mites by up to a 20% decrease in presence of HZ mullein bugs compared to the control treatment in the free-ranging test (and 46% in the sleeve cages). These results suggest that the control of spider mites below economic thresholds (about 7 to 15 spider mites per leaf) solely by mullein bugs would require a substantial number of individuals. The bugs though may have been introduced when the spider mites were already too numerous, as no intervention threshold was established before the experiment. Under these conditions, mullein bugs, even from a highly zoophagous lines, are not efficient enough to be used as a biological control agent. Rather, our results suggest that the role of mullein bugs in apple orchards is complementary to the effect of other predators, such as predatory mites.

Risks associated with zoophytophagous predators increase when their density is high relative to the density of their prey [32,33]. However, the level of damage to fruitlets was very low in our experiment, even if mullein bug nymphs did not have access to prey. Both lines generated low levels of punctures to apple fruitlets under our field conditions. Here, our results contrast with those of Kain and Agnello [19]. Kain and Agnello [19] observed damage in 87% of clusters when young mullein bug nymphs were introduced at bloom, whereas we observed that the percentage of damaged clusters was ~56% in the LZ treatment and ~37% in the HZ treatment. These contrasting results are consistent with the hypothesis that the difference in the status of mullein bugs from one region to another is related to a variation in the composition of the population [23]. In Quebec, some authors consider mullein bugs as predators that can provide some benefits in apple orchards [9,15], whereas in New York State, they are clearly identified as a pest (A. Agnello, personal communication). These variations between areas suggest that agricultural practices (e.g., use of broad-spectrum pesticides) or ecological factors (e.g., availability of alternative herbaceous hosts near orchards) may have generated a variation in the composition of mullein bug population in terms of individuals using a zoophytophagous or a phytozoophagous strategy [23].

Taken together, the strain selection did contribute to increasing the ratio of benefits/risks associated with zoophytophagous mullein bugs in an apple orchard. The mullein bug does not pose a risk to apple production once the fruitlets reach a size greater than 13 mm [19,20]. After this critical period, the mullein bug contributes to the control of spider mites. Our results indicate that only the HZ line significantly reduces the populations of spider mites. The conservation of such lineages in an agricultural setting would be desirable. On the other hand, lowly-zoophagous lines would have little positive effect while retaining the potential negative effects. Crop management has a great influence on genetic variability and selection and also on characteristics of local pest populations [34,35]. The evolution of organisms in agricultural environments can occur on an ecological scale [36,37,38,39]. This evolution, which can act very quickly under the influence of anthropic action, can no longer be neglected by pest management plans [36,38,39,40,41]. Lankau et al. [36] argue that evolutionary changes must be of central concern in conservation and pest management. Evolution can be concretely applied using approaches that influence selection, genetic variability, connectivity, and gene flow [36]. Any agricultural practice is likely to cause evolutionary changes in pests, but also in natural enemy populations [35,42], which subsequently may affect their ecological and economic values. The effect of both ecological and anthropogenic factors affecting the composition of mullein bug populations needs to be investigated in order to obtain long-term success either in natural control of spider mites or in a biological control program against spider mites. The adoption of selective methods on mullein bug populations would both maintain populations in adequate proportions and create benefits from the predatory action of mullein bugs.

Our results support the idea that genetic variation within zoophytophagous predatory populations should be considered in biological control based on this type of predator. The role of zoophytophagous predators has been studied extensively in tomato crops. *Macrolophus pygmaeus* Rambur (Hemiptera: Miridae) and *N. tenuis* are commercially available as biological control agents of whiteflies and South American tomato pinworms [31]. Commercializing genetically improved lines could improve this approach [25,43]. For instance, Nachappa et al. [44] observed that genetic variation in prey consumption, conversion efficiency, and dispersal in the specialist predatory mite *Phytoseiulus persimilis* (Athias-Henriot) (Acarina: Phytoseiidae) affect predator-prey interactions, long-term population dynamics, as well as efficiency in the biological control of two-spotted spider mites. Genetic improvement of predators used as biological control agents has earned scant attention, despite the great potential of considering the genetic aspects [25,43,45,46]. Our study demonstrated the potential of genetic improvement of predators in attributes that affect success in biological control (see also Nachappa et al. [44]). In the specific case of zoophytophagous predators, genetic improvement could target zoophagy and phytophagy simultaneously in order to increase the benefits and reduce the risks associated with these predators.

## 5. Conclusions

Our results indicate that both lines did not differ in the level of damage to apple fruits, but the highly-zoophagous line had a stronger impact on spider mite populations both in sleeve and free-ranging tests. However, mullein bugs did not maintain the pest population under the economic thresholds. We suggested that manipulating the composition of zoophytophagous predatory populations could increase benefits and/or reduce risks related to these predators in agroecosystems. However, managing the genetic pool of local populations could be more efficient than releasing reared individuals.

## Figures and Tables

**Figure 1 insects-10-00303-f001:**
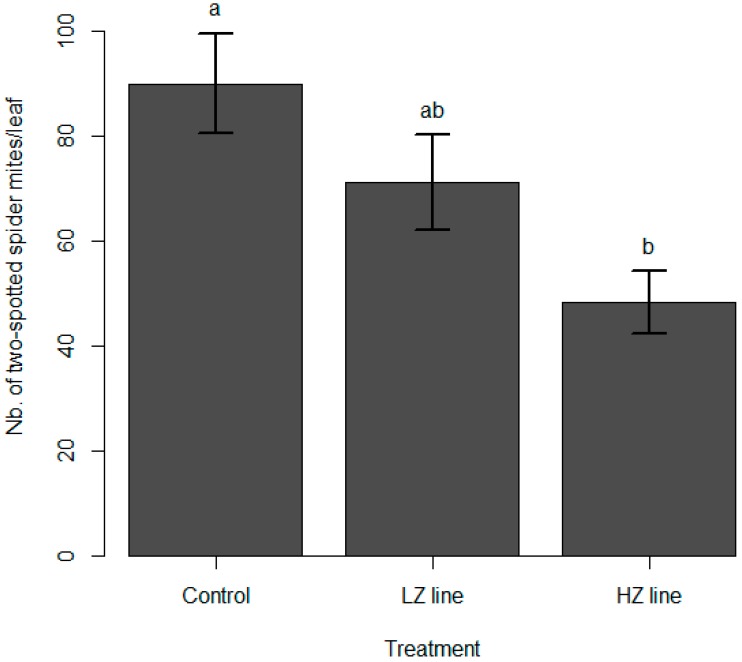
Number of two-spotted spider mites 14 days after the introduction of three mullein bug nymphs (N3 to N5) in sleeve cages. LZ line refers to the lowly-zoophagous isogroup line, whereas HZ indicates the highly-zoophagous line. In the control treatment, no mullein bug nymphs were introduced. Means with the same letter are not statistically different. Error bars indicate standard error.

**Figure 2 insects-10-00303-f002:**
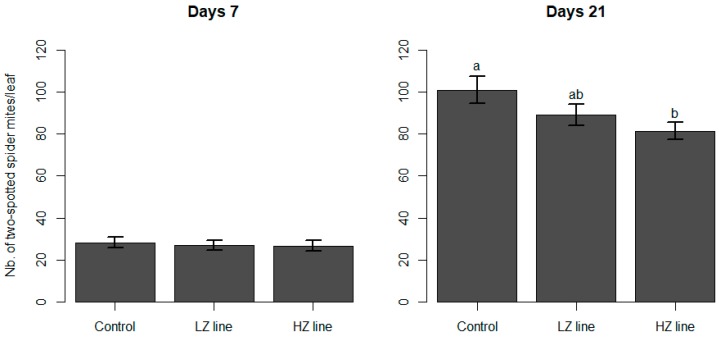
Number of two-spotted spider mites 7 and 21 days after the introduction of four free-ranging mullein bug nymphs (N3 to N5). LZ line refers to the lowly zoophagous isogroup line, whereas HZ indicates the highly-zoophagous line. In the control treatment, no mullein bug nymphs were introduced. Means with the same letter are not statistically different. Error bars indicate standard error.

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
