# Peer review of "Can Isogroup Selection of Highly Zoophagous Lines of a Zoophytophagous Bug Improve Biocontrol of Spider Mites in Apple Orchards?"

_insects, 2019, doi:10.3390/insects10090303_

Round 1
Reviewer 1 Report
The study reports on an interesting issue -selecting a population that is both phytophagous and zoophagous to increase pest control contribution. There is a great deal of interest in increasing understanding of the contribution made by zoophagous organisms to pest control and understanding the importance of bugs with dual roles has potential to contribute to increased pest control. Raises the issue of fully trying to understand impacts of management practices not only on pests but also on their control organisms. The inclusion of a field test to support for lab and semi field findings and the demonstration of no increased damage from the mullein bugs are both interesting.
L28 surrogate. Is this the best word?
L40 and elsewhere damages plural-question if appropriate
L42 notorious predators? Maybe replace with well known
L43 move (Acari: Tetranychidae) to follow (Koch) (L44)
L83repetition of detail for two-spotted spider mite unnecessary as previously given in introduction.
L93 using the method outlined in Dumont et al 2016
L94 and elsewhere P not p
L102, 103 if details of potato plants and soybeans need to be given they should be placed at first mention (L82)
L110 located ? Why localised when previously located used?
L117 trees
L140 fourteen for start of sentence
L157, L176 damage
L224-226 lacks clarity
Refs edit for consistency-eg capitalization of titles, titles in full/abbreviated
L322 and elsewhere Italicize species names
Fig captions insert Error bars represent …..
Author Response
L28 surrogate. Is this the best word?
** We change the word for substitute.
L40 and elsewhere damages plural-question if appropriate
** Ok.
L42 notorious predators? Maybe replace with well known.
** Done
L43 move (Acari: Tetranychidae) to follow (Koch) (L44)
** Done.
L83repetition of detail for two-spotted spider mite unnecessary as previously given in introduction.
** We removed it.
L93 using the method outlined in Dumont et al 2016
** Changed.
L94 and elsewhere P not p
** Ok.
L102, 103 if details of potato plants and soybeans need to be given they should be placed at first mention (L82)
** Fixed.
L110 located? Why localised when previously located used?
** Fixed.
L117 trees
** Fixed.
L140 fourteen for start of sentence
** Changed.
L157, L176 damage
** Corrected.
L224-226 lacks clarity
** Edited.
Reviewer 2 Report
Overall, this is a well-conducted study that describes differences in damage and biological control by a generalist predator that is known to both control a pest and cause crop damage. The discussion and introduction are well organized.
I have made some minor comments throughout.
The only major change needed is substantial English language editing for phrasing, tense, subject/verb agreement, etc.

Author Response
L58-59 A couple sentences summarizing the genetic differences would be helpful here.
** Line 65 to 69 provided more details on the genetic differences observed in previous works. We provided extra details in line 62 to 64.
L84 list the species of pollen.
***It’s not specified on the bottle. It’s just mixed flower pollen and it does not make a big difference here.
L93 Is this from the Dumont et al 2016 paper or a study being reported in this paper? If the prior, make that clear by removing the statistics and just citing the paper again. If the latter, more details on how the test was performed (sample size, methods) will need to be provided.
***It is from Dumont et al. 2016. As suggested by the reviewer, we removed the statistics and cited the paper.
Reviewer 3 Report
Please conduct a major revision of the manuscript to primarily correct the English text. Also, the formatting of the References is not consistent. Scientific names are customarily italicized in the text and Reference section.
Author Response
Please conduct a major revision of the manuscript to primarily correct the English text. Also, the formatting of the References is not consistent. Scientific names are customarily italicized in the text and Reference section.
*References were formatting properly and English text was revised.
Reviewer 4 Report
There are a few grammatical errors that I have pointed out on the attached PDF version of the manuscript.
One question regarding the statistics. Did the authors transform the proportion data prior to analysis?
I enjoyed reading the discussion.

Author Response
Dear Editor,
Please find attached a copy of the revised manuscript "Can isogroup selection of highly zoophagous lines of a zoophytophagous bug improve biocontrol of spider mites in apple orchards?". We would like to thank you and the anonymous referees for the constructive comments. We hope that the revisions have clarified the points raised by the referees. Below we explain the changes we have made following the referees’ suggestions and reply to their comments.
Reviewer 1
L28 surrogate. Is this the best word?
** We change the word for substitute.
L40 and elsewhere damages plural-question if appropriate
** Ok.
L42 notorious predators? Maybe replace with well known.
** Done
L43 move (Acari: Tetranychidae) to follow (Koch) (L44)
** Done.
L83repetition of detail for two-spotted spider mite unnecessary as previously given in the introduction.
** We removed it.
L93 using the method outlined in Dumont et al 2016
** Changed.
L94 and elsewhere P not p
** Ok.
L102, 103 if details of potato plants and soybeans need to be given they should be placed at first mention (L82)
** Fixed.
L110 located? Why localised when previously located used?
** Fixed.
L117 trees
** Fixed.
L140 fourteen for start of sentence
** Changed.
L157, L176 damage
** Corrected.
L224-226 lacks clarity
** Edited.
Reviewer 2
L58-59 A couple sentences summarizing the genetic differences would be helpful here.
** Line 65 to 69 provided more details on the genetic differences observed in previous works. We provided extra details in line 62 to 64.
L84 list the species of pollen.
***It’s not specified on the bottle. It’s just mixed flower pollen and it does not make a big difference here.
L93 Is this from the Dumont et al 2016 paper or a study being reported in this paper? If the prior, make that clear by removing the statistics and just citing the paper again. If the latter, more details on how the test was performed (sample size, methods) will need to be provided.
***It is from Dumont et al. 2016. As suggested by the reviewer, we removed the statistics and cited the paper.
Reviewer 3
Please conduct a major revision of the manuscript to primarily correct the English text. Also, the formatting of the References is not consistent. Scientific names are customarily italicized in the text and Reference section.
*References were formatting properly and English text was revised.
Reviewer 4
L179-180 Since proportions were analyzed, were the data transformed?
** We used generalized linear models for binomial/proportional distributed data. Hence, data transformation was not useful.
L186-187 That is a lot of mites per leaf. Was there not leaf drop?
** We did not measure leaf drop. However, we did not notice such a thing.
L225-227 I am confused. If I read correctly the authors stated above that the HZ line had a significant impact on spider mite populations and LZ did not. In this sentence, they state no difference in SM population.
** HZ line is different from the control treatment, but not LZ. However, HZ and LZ are not statistically different. Hence, LZ has a mild impact on SM, which lead to an intermediate result between the control and the HZ treatments. On Figure 2, you can notice that the control treatment was attributed the letter A and the HZ line got the letter B. LZ line is in between with the letter AB.
L249-251 Are there pesticide use data that could support this statement?
** Not to our knowledge. It will be very interesting though.